# Development of Direct Immobilization Technique of Ag Nanoparticles on Resin Substrates Imparting High Antibacterial and Antiviral Activities

**DOI:** 10.3390/nano12173046

**Published:** 2022-09-02

**Authors:** Satoshi Seino, Yuji Ohkubo, Tomonari Magara, Hiroki Enomoto, Eri Nakajima, Tomoki Nishida, Yasuo Imoto, Takashi Nakagawa

**Affiliations:** 1Graduate School of Engineering, Osaka University, 2-1 Yamadaoka, Suita 565-0871, Japan; 2Japan Textile Products Quality and Technology Center, 5-7-3, Shimoyamate-dori, Chuo-ku, Kobe-City 650-0011, Japan

**Keywords:** silver, nanoparticle, resin, antibacterial, antiviral

## Abstract

A new method has been developed to impart the antimicrobial activity of silver nanoparticles to resin substrates. A resin substrate immersed in an aqueous solution of silver nitrate was irradiated with gamma ray or high energy electron beams. Silver nanoparticles were successfully immobilized on the resin surface directly by chemical reactions induced by ionizing radiation. It was experimentally confirmed that various resin materials, such as acrylonitrile-butadiene-styrene, polyethylene, polypropylene, polyvinyl chloride, and polycarbonate, were applicable for this process. The effects of gamma ray or electron beam irradiation on resin substrates were almost negligible since the irradiation dose was equal or less than that used for sterilization. Despite the small amount of Ag loadings, the obtained samples showed high antibacterial and antiviral activities.

## 1. Introduction

Antimicrobial technology is attracting much attention to improve the safety of living environments. Organic antimicrobial agents, such as alcohols and hypochlorous acids, are widely used for the purpose of disinfecting hands and familiar objects. On the other hand, inorganic antimicrobial agents are usually used to impart antimicrobial activities to base substrates, such as resin, fibers, and so on. Conventional methods for immobilizing antimicrobial agents on resin substrates include a kneading method or a coating method. In the kneading method, antimicrobial agents are mixed with the resin’s raw materials during manufacturing processes. In the coating method, antimicrobial agents are mixed with coating materials and/or binders and then applied to the resin surfaces. Since the antimicrobial agents are either buried inside the resin or completely covered with coating materials and/or binders, they tend not to work, thus excess amounts of additional antimicrobial agents are sometimes required to obtain the required performance.

Among various kinds of inorganic antimicrobial agents, silver nanoparticles have attracted a great deal of attention because of their excellent antimicrobial ability and chemical stability [1,2,3]. However, conventional immobilizing methods are difficult to be applied because their sizes are too small. Our research group succeeded in immobilizing metallic Ag nanoparticles on textile fabrics using nanoparticle synthesis techniques in an aqueous solution system using radiation [4]. Chemical reactions induced by ionizing radiation reduce Ag+ to form metallic Ag nanoparticles, which were directly immobilized onto support textile fabrics without any binders. Thus the obtained Ag nanoparticles on textile fabrics showed high antimicrobial activities against bacteria [4], molds [5], and viruses [6]. In this paper, we report the direct immobilization technique of Ag nanoparticles on resin substrates using the same technique with radiation. When compared with textile fabrics, the immobilization of Ag nanoparticles on resin plates is supposed to be difficult because of their lower specific surface area and wettability. However, our research group succeeded in immobilizing Pt nanoparticles on ABS substrates by using the presented procedure with electron beam irradiation [7]. We expected that similar results would be obtained with Ag nanoparticles. Support resin plates were immersed in aqueous solutions of AgNO_3_, to which gamma rays or high-energy electron beams were irradiated. The antimicrobial activity of the resin plate modified with Ag nanoparticles were evaluated. The schematic illustration of this work is presented in Appendix A.

## 2. Materials and Methods

### 2.1. Material Characterization of Ag on ABS

The chemical reagents of AgNO_3_ and 2-propanol were purchased from Wako Pure Chemical Industries Ltd. (Osaka, Japan). Pure water was obtained using a Millipore Direct-Q system (Merck Millipore, Burlington, MA, USA). The resin substrates used in this study were commercial plates of ABS (acrylonitrile-butadiene-styrene), PE (polyethylene), PP (polypropylene), PVC (polyvinyl chloride), and PC (polycarbonate). The size of each resin plate was 50 mm × 50 mm × 3 mm. All plates were prepared for use by washing with pure water. Plastic zipper bags made of polyethylene were used as reaction vessels.

### 2.2. Radiochemical Synthesis of Silver Nanoparticles on Resin Substrates

To provide a silver source, aqueous solutions were prepared by dissolving AgNO_3_ powder in pure water at concentrations of 1.0 mM. Next, 2-propanol was added to the solution as a radical scavenger with concentrations of 0%, 1%, and 10% vol. Samples of a resin plate were immersed in 20 mL of precursor solution and then closed in a plastic bag. The samples were irradiated with ^60^Co gamma ray or high-energy electron beam at room temperature. Gamma ray irradiation was carried out at KOGA ISOTOPE Ltd. (Shiga, Japan) and was controlled to deliver an absorbed dose of 20 kGy. Electron beam irradiation was carried out at the SHI-ATEX Co., Ltd. (Osaka, Japan), using a 4.8 MeV beam to deliver a controlled surface dose of 20 kGy. After the irradiation, resin plates were washed with pure water several times and then air-dried at room temperature.

### 2.3. Material Characterization

Observation of Ag nanoparticles immobilized on the resin plate was performed by SEM (JSM-7001F, JEOL Ltd., Tokyo, Japan) using an acceleration voltage of 10 kV. Before the observation, resin plates were treated with Osmium Plasma Coater (OPC60A, Filgen, Aichi, Japan) to ensure conductivity. The amount of Ag nanoparticles immobilized on each resin plate was analyzed by dissolving them in HNO_3_ solution (ca. 10%) using an ICP-AES (ICPE-9000, Shimadzu Corp., Kyoto, Japan). The chemical state of Ag on the resin plate was investigated by XPS (Al-*K*_α_, 15 kV, Quantum 2000, ULVAC-PHI, Inc., Kanagawa, Japan), and XPS spectra were analyzed by the XPSpeak41 program to determine peak positions. The binding energy for each spectrum were discussed based on database [8].

### 2.4. Antibacterial Test

The antibacterial activity was investigated against gram-negative *Escherichia coli* (NBRC 3972). Antibacterial tests were performed using the film adhesion method in essentially the same manner outlined in JIS Z 2801. The detailed procedure is as follows:Before the test, the surface of the resin plate was wiped with gauze that had absorbed ethanol at a purity greater than 99%, and then air-dried at room temperature.The bacterial solution (0.4 mL) was added dropwise to a resin plate of 50 mm × 50 mm.The surface of the resin plate was covered with a 40 mm × 40 mm polyethylene film to spread the bacterial solution evenly over the resin plate.Samples were incubated at 35 ± 1 °C and a relative humidity of no lower than 90% for 24 h.After incubation, 10 mL of SCDLP medium was used to wash out the bacterial solution.A series of 10-fold dilutions of washed-out suspension was prepared by using phosphate-buffered physiological saline. Next, 1 mL of each dilution was placed into separate sterile Petri dishes, and 15 mL of plate count agar (E-KB07, EIKEN CHEMICAL Co., Ltd., Tokyo, Japan) was poured into each Petri dish and swirled gently to disperse the bacteria. Then, the Petri dishes were incubated at 35 ± 1 °C for 40–48 h.Viable bacteria count per 1.0 mL of washed-out suspensions was measured using the plate count method. The results describe the viable bacteria count per cm^2^ of resin plate.

### 2.5. Antiviral Test

The antiviral activity was investigated against the influenza A virus (H3N2, A/Hong Kong/8/68; TC adapted ATCC VR-1679) and the SARS-CoV-2 (JPN/TY/WK-521, Distributed from National Institute of Infectious Diseases, Tokyo, Japan). Antibacterial tests were performed using the film adhesion method in essentially the same manner outlined in ISO 21702. The host cell lines were MDCK cells ATCC CCL-34 for the influenza A virus, and VeroE6/TMPRSS2 (JCRB1819) for the SARS-CoV-2 virus. The detailed procedure is as follows:Before the test, the surface of the resin plate was wiped with gauze that had absorbed ethanol at a purity greater than 99% and then air-dried at room temperature.The viral solution (0.4 mL) was added dropwise to a resin plate of 50 mm × 50 mm.The surface of the resin plate was covered with a 40 mm × 40 mm polyethylene film to spread the viral solution evenly over the resin plate.Samples were incubated at 25 ± 1 °C and a relative humidity of no lower than 90% for 24 h.After incubation, 10 mL of SCDLP medium was used to wash out the viral solution.A series of 10-fold dilutions of washed-out virus solution was prepared by using EMEM (for Influenza A virus) or 2% FBS-containing DMEM (for SARS-CoV-2). The growth medium was removed from 6-well plastic plates with the host cell monolayer, and inoculated in 0.1 mL of each solution. Next, the plates were placed in the CO_2_ incubator at 34 °C (for Influenza A virus) or 37 °C (for SARS-CoV-2) for the chosen time. After incubating for chosen time, 3 mL of the agar medium was poured into the wells for plaque assay. Then, the plates were incubated at 34 °C (for Influenza A virus) or 37 °C (for SARS-CoV-2) in the CO_2_ incubator for 2 to 3 days.The viral infectivity titer per 0.1 mL of washed-out virus solution was measured by plaque assay and calculate the viral infectivity titer per cm^2^ of resin plate.

## 3. Results

### 3.1. Material Characterization of Ag on ABS

The appearance of the resin plate was visually evaluated before and after the Ag immobilization process. Transparent resin plates, such as PP and PC, were slightly colored light yellow, which was ascribed to the surface plasmon resonance of metallic Ag nanoparticles. On the other hand, the color change was relatively inconspicuous in the case of the opaque resins (ABS, PE, and PVC). Figure 1 shows SEM images of Ag/ABS samples. SEM-EDX analysis revealed that the small aggregates seen as white spots consisted of Ag. With electron beam irradiation, Ag particles with several tens of nanometers formed larger aggregates (Appendix A). In the case of gamma ray irradiation, relatively smaller Ag nanoparticles appeared to be well dispersed on the ABS surface. From the SEM observation, it was difficult to distinguish the damage to the resin surface due to irradiation.

The existence of Ag on the resin plates’ surface was confirmed by ICP-AES analysis. However, experimental data for the loading amounts showed statistical dispersion around the lower limit of quantification, approximately 0.1 μg-Ag/cm^2^. Therefore, it was difficult to discuss the effects of the experimental parameters on Ag loading amounts.

Figure 2 shows the Ag3d-, C1s- and O1s-XPS spectra for the Ag/ABS samples. XPS peaks of Ag3d_5/2_ and Ag3d_3/2_ were observed in all samples, indicating the existence of Ag. The low S/N ratio of the spectrum is attributed to the small amount of Ag loadings. O1s peaks were small, even with the control ABS substrate. Since oxygen is not contained in the composition formula of ABS, the observed O1s peak is assumed to be mainly due to the contamination and/or impurities. Peak positions for each spectrum are summarized in Table 1. It has been reported that Ag3d_5/2_ has binding energies for Ag metal, Ag_2_O and AgO are 368.2, 367.9 and 367.6 eV, respectively [8]. The peak positions of the present Ag/ABS samples were between the Ag metal and Ag oxides. As the XPS analysis is sensitive to surface composition, surfaces of Ag nanoparticles on ABS are considered to be partially oxidized. Small O1s peaks were observed with binding energies of 531.7–532.1 eV, which is close to the value of C=O (531.9 eV) and C–O (531.8 eV). The O1s peak for Ag_2_O (530.1 eV) was not observed, probably due to smaller amounts of Ag oxides. The peak position of 284.6 eV for C1s is attributed to the C–H and/or C–C bonding, which is attributed to the compositions of the ABS substrate. The present XPS results for C1s and O1s indicate that radiolytic surface modifications of ABS were almost negligible for the present irradiation conditions.

### 3.2. Antibacterial Activity

Table 2 shows the antibacterial activity of Ag/ABS samples against *E. coli*. It was confirmed that the support ABS substrate used in this study has no antibacterial activity against *E. coli*. Ag/ABS samples were prepared by using gamma ray or electron beam irradiation in the absence of 2-propanol. Both Ag/ABS samples showed excellent antibacterial activity, with the viable bacteria count drastically decreasing to below detection level.

Table 3 shows the antibacterial activity of Ag nanoparticles with support resin plates of PE, PP, PVC and PC. The samples were prepared by using gamma ray irradiation with 2-propanol concentrations of 0 or 10% vol. In all cases, the Ag-immobilized resin plates demonstrated excellent antibacterial activities. These results indicate that the present Ag immobilization technique is applicable to various resin substrates.

### 3.3. Antiviral Activity

Antiviral activities of Ag/ABS samples were then examined. Ag/ABS samples prepared by using electron beam with 2-propanol concentrations of 0%, 1%, and 10% vol. were tested against influenza-A virus. As shown in Table 4, all the Ag/ABS samples showed antiviral activity against influenza-A virus. Although some variations were observed between three tested samples (Appendix A), the antiviral activity depended on the synthesis conditions; higher 2-propanol concentrations resulted in higher antiviral activity. This difference might be caused by the loading amount of Ag, as implied by the XPS Ag3d peak intensity shown in Figure 2a. For further discussion, the immobilization technique must be improved to control the conditions of Ag nanoparticles on resin substrates, such as size, chemical state, loading amount and dispersibility on substrates.

Table 5 shows the antiviral activities of Ag/ABS samples against SARS-CoV-2. The Ag/ABS samples were prepared by using gamma ray irradiation in the absence of 2-propanol. The result clearly indicated that Ag nanoparticles on ABS are also effective against the coronavirus. These results indicate that the presented Ag immobilization technique could contribute to preventing infectious diseases.

## 4. Discussion

Immobilization processes of Ag nanoparticles on resin plates are now discussed. The radiochemical processes for the reduction of Ag ions in an aqueous solution system has been previously described as follows [9]:(1)$$H_{2}O\to e_{aq}^{-}{,\;H}^{•}{,\;\mathrm{OH}}^{•},\;\mathrm{etc}$$
(2)$${\left({\mathrm{CH}}_{3}\right)}_{2}\mathrm{CHOH}+{\mathrm{OH}}^{•}\to {\left({\mathrm{CH}}_{3}\right)}_{2}^{•}{\mathrm{COH}}_{}+H_{2}O$$
(3)$${\left({\mathrm{CH}}_{3}\right)}_{2}\mathrm{CHOH}+H^{•}\to_{}{\left({\mathrm{CH}}_{3}\right)}_{2}^{•}\mathrm{COH}+H_{2}$$
(4)$${\mathrm{Ag}}^{+}+e_{aq}^{-}\to \mathrm{Ag}$$
(5)$${\mathrm{Ag}}^{+}+H^{•}\to \mathrm{Ag}+H^{+}$$
(6)$${\mathrm{Ag}}^{+}+{\left({\mathrm{CH}}_{3}\right)}_{2}^{•}\mathrm{COH}\to \mathrm{Ag}+{\left({\mathrm{CH}}_{3}\right)}_{2}\mathrm{CO}$$

Note that Equation (1) represents water radiolysis, and that the hydrogen atoms and hydrated electrons generated are strong reductants capable of reducing Ag^+^ to metallic Ag^0^. In the presence of 2-propanol, the radicals of H^•^ and OH^•^ are scavenged to form alcohol radicals, in accordance with the Equations (2) and (3), and the generated radicals also contribute to the reduction of Ag^+^. Metallic Ag nuclei grow up in aqueous solutions to form metallic Ag nanoparticles, which are immobilized on the surface of the support resin as a way of minimizing their surface energy. As shown in the XPS analysis, Ag on ABS substrate were partially oxidized, which are assumed to be surface oxidization of nanoparticles by the surrounding atmosphere. It is also necessary to consider the effect of irradiation on the resin substrates, as ionizing radiation could induce direct and/or indirect action on polymers. However, as shown in XPS analysis data, the chemical change on the resin surface seems to be almost negligible under the presented irradiation conditions. Since the irradiation dose is equal or less than that used for the sterilization of medical equipment, deterioration of the resin substrates scarcely proceeded [10].

It should be noted that Ag nanoparticles immobilized on resin plates showed high antibacterial and antiviral activities despite their low amounts of loadings. With conventional methods, it was necessary to add 0.5 to 1% wt of silver nanoparticles to the resin substrate to impart sufficient antimicrobial activities [11,12]. On the other hand, Ag nanoparticles loaded on resin plates obtained in this work were less than 0.001% wt. These comparisons clearly indicate the superiority of the presented immobilization technique over the conventional methods. As all the silver nanoparticles exist only on the resin surface, they can be directly contacted with bacteria and viruses. The antimicrobial mechanism of silver nanoparticles have been discussed in previous studies. It has been reported that Ag^+^ ion release is a major pathway for the antimicrobial activity of Ag nanoparticles, and the peroxidation of Ag nanoparticle surfaces enhances the ion release rate constant [3]. Surface oxidation of Ag nanoparticles immobilized on resin substrates might also contributed to the presented high antimicrobial activities.

## 5. Conclusions

We have succeeded in immobilizing Ag nanoparticles on resin surfaces directly by chemical reactions induced by ionizing radiation. The resin substrates showed high antibacterial and antiviral activities even with extremely low amounts of Ag loadings. Although high antimicrobial activities were successfully imparted to resin substrates, there are some points still to be improved. In the ICP-AES analysis, there was a non-negligible variation in numerical values for Ag loading amounts even with the same synthesis conditions. The distribution of silver nanoparticles on the resin surfaces was not uniform, as shown in the SEM images. Further studies are required to improve this immobilization technique. We expect this technique to not only be applied for the imparting of antibacterial property to daily necessaries, but also for preventing infectious diseases caused by contact infection.

## Figures and Tables

**Figure 1 nanomaterials-12-03046-f001:**
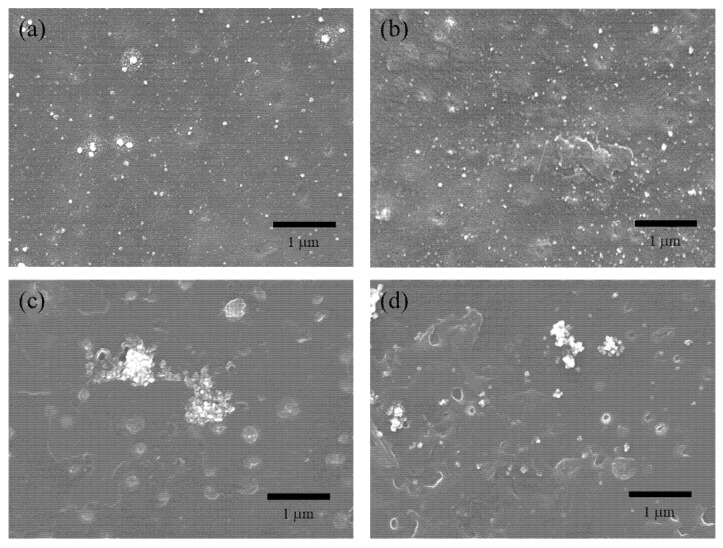
SEM images of Ag/ABS samples synthesized by using radiation. (**a**) Gamma ray, without 2-prop.; (**b**) Gamma ray, with 2-prop. (10% vol.); (**c**) Electron beam, without 2-prop.; (**d**) Electron beam with 2-prop (10% vol.).

**Figure 2 nanomaterials-12-03046-f002:**
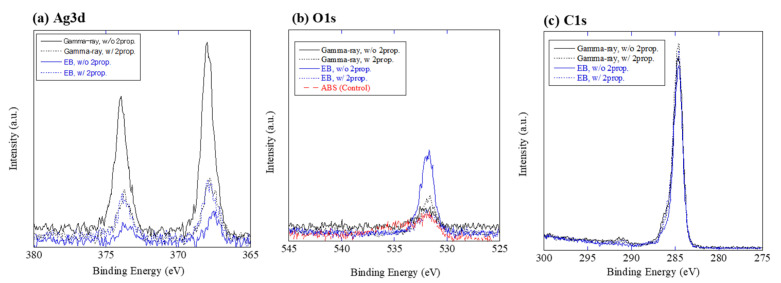
XPS spectra of (**a**) Ag3d, (**b**) C1s and (**c**) O1s for Ag/ABS samples.

**Table 1 nanomaterials-12-03046-t001:** Peak positions of the Ag3d-, C1s- and O1s-XPS spectra for Ag/ABS samples.

Sample ID	XPS Peak Energy		
	Ag3d_5/2_	O1s	C1s
Gamma ray, without 2-prop.	368.0 eV	532.1 eV	284.6 eV
Gamma ray, with 2-prop. (10% vol.)	367.8 eV	531.7 eV	284.6 eV
Electron Beam, without 2-prop.	367.6 eV	531.9 eV	284.6 eV
Electron Beam, with 2-prop. (10% vol.)	367.8 eV	531.9 eV	284.6 eV

**Table 2 nanomaterials-12-03046-t002:** Antibacterial activity of Ag/ABS samples against *E. coli*.

Samples		Viable Cell Count *^1^
PE Film (Control)	Immediately after inoculation	4.05
	After 24 h	6.08
ABS (Control)	After 24 h	5.69
Ag/ABS (Gamma ray, without 2-prop.)	After 24 h	<−0.20
Ag/ABS (Electron Beam, without 2-prop.)	After 24 h	<−0.20

*^1^ The average common logarithm for the number of bacterial colonies obtained from three test samples. Original data is shown in Appendix A.

**Table 3 nanomaterials-12-03046-t003:** Antibacterial activity of Ag/resin-plate samples prepared by gamma ray against *E. coli*.

Samples		Viable Cell Count *^1^
PE Film (Control)	Immediately after inoculation	4.05
	After 24 h	5.86
PE (Control)	After 24 h	4.80
Ag/PE (without 2-prop.)	After 24 h	<−0.20
Ag/PE (with 2-prop. 10% vol.)	After 24 h	<−0.20
PP (Control)	After 24 h	3.11
Ag/PP (without 2-prop.)	After 24 h	<−0.20
Ag/PP (with 2-prop. 10% vol.)	After 24 h	<−0.20
PVC (Control)	After 24 h	5.55
Ag/PVC (without 2-prop.)	After 24 h	<−0.20
Ag/PVC (with 2-prop. 10% vol.)	After 24 h	<−0.20
PC (Control)	After 24 h	5.64
Ag/PC (without 2-prop.)	After 24 h	<−0.20
Ag/PC (with 2-prop. 10% vol.)	After 24 h	<−0.20

*^1^ The average common logarithm for the number of bacterial colonies obtained from three test samples. Original data is shown in Appendix A.

**Table 4 nanomaterials-12-03046-t004:** Antiviral activity of Ag/ABS samples prepared by electron beam against Influenza-A virus.

Samples		Common Logarithm of Infectivity Titer Value (PFU/cm^2^) *^1^
ABS (Control)	Immediately after inoculation	5.46
	After 24 h	4.63
Ag/ABS (EB, without 2-prop.)	After 24 h	2.88
Ag/ABS (EB, with 2-prop. 1% vol.)	After 24 h	2.66
Ag/ABS (EB, with 2-prop. 10% vol.)	After 24 h	1.53

Test Virus; Influenza A virus (H3N2) (A/Hong Kong/8/68; TC adapted ATCC 1679). *^1^ The average common logarithm of infectivity titer value obtained from three test samples. Original data is shown in Appendix A.

**Table 5 nanomaterials-12-03046-t005:** Antiviral activity of Ag/ABS samples prepared by gamma ray against SARS-CoV-2.

Samples		Common Logarithm of Infectivity Titer Value (PFU/cm^2^) *^1^
ABS (Control)	Immediately after inoculation	5.65
	After 24 h	4.82
Ag/ABS (Gamma ray, without 2-prop.)	After 24 h	1.06

Test Virus; SARS-CoV-2, (NIID isolate: JPN/TY/WK-521). *^1^ The average common logarithm of infectivity titer value obtained from three test samples. Original data is shown in Appendix A.

## Data Availability

Not applicable.

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
