# Peer review of "Development of Direct Immobilization Technique of Ag Nanoparticles on Resin Substrates Imparting High Antibacterial and Antiviral Activities"

_nanomaterials, 2022, doi:10.3390/nano12173046_

Round 1

Reviewer 1 Report

A method for fabrication of resin substrates containing embedded silver nanoparticles and possessing antimicrobial and antiviral activities is proposed. The method is based on irradiation of resins immersed in aqueous solution of silver nitrate with γ-ray or electron beam. Since antimicrobial technology is an important tool for protecting the living environment, the aim of the presented research is of obvious importance. However, to be worthy of publication, the communication requires minor corrections in accordance with the following comments:

1.     General comment. Did the author evaluated the possibility to impregnate the same resins with Ag nanoparticles by their incubation in silver nitrate solution followed by addition of a proper reducing agent (wet chemistry method)? Such an approach does not require sophisticated equipment and can be easily performed. 

2.     Page 2, line 57, page 5, line 185, page 6, lines 198, 201, 206, and page 7, line224: should be 2-propanol instead of 2-Propanol.

3.     Page 2, line 67: should be ‘were” instead of “was”.

4.     Page 3, items 6 and 7 of the detailed procedure as well as 6 and 7 of the antiviral test: I suppose it should be not “prepare”, but “was prepared”, not “measure”, but “was measured”. The corresponding sentences should be corrected.

5.     Page 7, two sentences in lines 224-228: In my opinion, the sentences should be “In the presence of 2-propanol, the radicals H and OH· are scavenged to form alcohol radicals according to equations (2) and (3), and generated radicals also contribute to the reduction of Ag+. Metallic nuclei grow up in aqueous medium to form Ag nanoparticles, which are immobilized on the surface of the support resin as a way of minimizing their surface energy”.

6.     Since the antiviral activity of the studied objects was also demonstrated, I recommend modifying the title as “Development of……. imparting high antimicrobial and antiviral activities.

Author Response

We wish to express our appreciation to the Reviewer for his or her insightful comments, which have helped us significantly improve the paper. We thank the Reviewer for this pertinent comment.  Please see the attachment to check our response.

Reviewer 2 Report

In this manuscript, Satoshi Seino et al developed a direct immobilization method to construct Ag nanoparticles on resin with high antimicrobial activity. The topic is interesting and the content is organized very well. Basically, I recommend it to be published in this journal after considering the following points.

 1. English needs to be improved seriously in the whole manuscript.

 2. It will be much better if the authors give a schematic illustration of this work.

 3. Importantly, the standard errors should be added in all Tables.

4. The authors should provide Ag content in the hybrid biomaterials using some techniques such as TGA.

 5.  To show the advantage of this method, the authors could add a comparison of present nanomaterials with antimicrobial activities.

 6. Some recent reports on immobilization of noble metal-based nanomaterials and their bioapplications are recommended to be cited, such as ACS Nano, 2014, 8, 8529-8536; Angew. Chem. Int. Ed. 2019, 58, 557-5576; Nanoscale 2016, 8, 12095-12104; ACS Sens. 2021, 6, 2290-2298.

Author Response

(The authors gave the same response as above.)

Reviewer 3 Report

- Antibacterial test should be carried out on G+ve bacteria

- That results of antibacterial for all the tested samples. Author should carry out another test for better comparison among the used resins

- The discussion is poor, need more explanations

- What do the author mean by (-0.20) for cell viable

- TEM for the AgNPs formed inside the resin should be carried out and the size of nano should be measured

-ICP details should be deleted as long as the author didn't provide  appropriate results and discussion

- XPS images didn't confirm the formation of AgNPs 

Author Response

(The authors gave the same response as above.)
